# A pathway to coexistence of electroluminescence and photovoltaic conversion in organic devices

Taku Oono [1], Yusuke Aoki[2], Tsubasa Sasaki[1], Haruto Shoji[2], Takuya Okada[1], Takahisa Shimizu [1,2], Takuji Hatakeyama [3] & Hirohiko Fukagawa [4] ✉

Achieving both high electroluminescence (EL) efficiency and power conversion efficiency (PCE) in a single organic device has long been considered difficult, since the design principles optimising one often compromise the other. In this study, we present a strategy employing multiple-resonance thermally activated delayed fluorescence materials with strong absorption and high emission efficiency, enabling coexistence of high EL and photovoltaic (PV) efficiencies. By precisely controlling charge-transfer states at donor/acceptor interfaces, we successfully achieve full-spectrum visible EL while maintaining efficient charge generation essential for PV operation. The optimised multifunctional devices exhibit emission colours ranging from blue to red, as well as white, with the green- and orange-light-emitting devices achieving an external quantum efficiency of EL exceeding 8.5% and a PCE of about 0.5%. These findings not only mitigate conventional efficiency trade-offs in organic devices but also open future avenues for emerging applications, including self-powered displays and lighting, potentially advancing optoelectronic technologies.

In recent years, the remarkable design flexibility of organic semiconductors has driven intensive research into a wide range of optical devices. Notable examples include organic light-emitting diodes (OLEDs), organic photovoltaics (OPVs), and various sensors, all of which have seen significant advancements in material development and performance enhancement[1–7]. The key factors governing the functionality of these light-emitting, energy-harvesting, and sensing devices are excitons and charge transfer (CT) states, which are bound states of an electron and a hole. In OLEDs, charge injection and transport lead to exciton/CT state formation, followed by radiative recombination that results in light emission[8–13]. Conversely, in OPVs and sensors, such as organic photodiodes (OPDs), the excitons/CT states generated by incident light must dissociate into free charges, which are subsequently transported and collected[14–17]. Therefore, precise control over exciton/CT state energy and dynamics holds the potential not only to enhance device performance but also to enable the development of dual-functional (DF) and multifunctional (MF) devices that integrate multiple functionalities, such as electroluminescence (EL) devices, photovoltaics (PVs), and PDs into a single platform[18–24]. Initially, many devices were reported as DF devices (EL devices and PVs). However, since PDs and PVs share similar operating mechanisms, DF devices can also function as MF devices (EL devices, PVs, and PDs). MF organic devices represent a groundbreaking technology with the potential to significantly contribute to both the advancement of an increasingly digitalized society and sustainable energy solutions. To enhance security in mobile displays, such as those in smartphones, OLED and OPD panels are being developed separately[25]. If a high-performance device integrating both EL and PD functions can be realised, there will no longer be a need for a separate OPD panel. Furthermore, if both EL and PV functions can be combined,

[1]Japan Broadcasting Corporation (NHK), Science & Technology Research Laboratories, Setagaya-ku, Tokyo, Japan. [2]Department of Applied Physics, Tokyo University of Science, Tokyo, Japan. [3]Department of Chemistry, School of Science, Kyoto University, Sakyo-ku, Kyoto, Japan. [4]Center for Frontier Science, Chiba University, Chiba-shi, Chiba, Japan. ✉e-mail: hiro.fukagawa@chiba-u.jp

it could enable mobile displays that recharge under indoor lighting or sunlight when not in use. While we note that the optimal PV device should ideally harvest part of the near-infrared spectrum of sunlight, which is difficult to exploit simultaneously for visible emission, the demonstration of efficient MF devices nonetheless suggests a pathway toward reducing charging needs and lowering the overall energy consumption of displays.

Despite several reports on the advances made on MF devices, achieving both high EL efficiency and power conversion efficiency ($PCE_{PV}$, defined as the ratio of generated electrical power to incident optical power in photovoltaic operation) remains a significant challenge (open symbols in Fig. 1). The first reported DF device was fabricated employing rubrene as the electron-donating molecule (donor) and fullerene ($C_{60}$) as the electron-accepting molecule (acceptor). Although this device exhibited a $PCE_{PV}$ of about 3%, its external quantum efficiency (EQE) of EL emission ($EQE_{EL}$) remained below 0.001%[18]. In this system, CT states are formed at the rubrene/$C_{60}$ interface with an energy ($E_{CT}$) of about 1.46 eV in the near-infrared region (~ 850 nm)[26]. By utilising triplet–triplet annihilation (TTA), we can realise EL emission in the orange region (~560 nm). To date, almost all reported organic MF devices have been based on rubrene[18–21]. However, the $EQE_{EL}$ in all reported devices with rubrene remains below 0.001%. One of the primary reasons for this low $EQE_{EL}$ is the limited efficiency of TTA-based upconversion (TTA-UC)[27]. Furthermore, a major limitation in enhancing PV performance arises from the high exciton binding energy ($E_b$) of CT states. Although the $E_b$ of CT states is considered lower than that of Frenkel excitons, which ranges from 0.2 to 1.2 eV, it often falls within the range of 0.2 to 0.6 eV[28–30]. A high $E_b$ is desired to facilitate charge recombination in EL devices while hindering charge separation in PVs[31,32]. Thus, in organic-semiconductor-based devices, it is inherently challenging to achieve high performance in both EL and PV functions, as the design principles that enhance one often compromise the other. Recently, an orange-light-emitting MF device fabricated without utilising TTA-UC has been reported; its $PCE_{PV}$ remains at approximately 0.8% and its $EQE_{EL}$ is about 1.5%[22,23]. The only reported MF device in which high $EQE_{EL}$ and $PCE_{PV}$ are both successfully realised utilises quantum dots (QDs), achieving an $EQE_{EL}$ of 8% and a $PCE_{PV}$ of approximately 0.2%[24]. However, the emission colour of QD-based devices is restricted to red. To realise advanced devices, such as self-powered displays and lighting, it is necessary to overcome the trade-off between $PCE_{PV}$ and $EQE_{EL}$ and develop MF devices that exhibit various emission colours.

Here, we present the strategy for both material selection and device design to achieve MF devices with both high $PCE_{PV}$ and $EQE_{EL}$, as well as the control of CT states to realise various emission colours ranging from blue to red. By using multiple-resonance thermally activated delayed fluorescence (MR-TADF) materials, which exhibit high emission efficiency and strong absorption, as the donor and acceptor materials with a low electron affinity (EA) of 2–3 eV, we successfully fabricated multiple MF devices exhibiting visible EL without utilising TTA-UC and a $PCE_{PV}$ exceeding 1%[33–35]. The $E_b$ of the CT state was found to depend on the donor–acceptor combination, ranging from approximately 0 to 0.4 eV. It was revealed that $E_b$ has a greater impact on the EL characteristics of MF devices than on their PV properties. Although the EL emission from the CT state (exciplex) is dominant in most devices, it was found that the emission colour can be tuned, as some devices exhibited EL from MR-TADF materials depending on the magnitude of $E_b$. On the basis of these insights, MF devices capable of emitting blue, green, yellow, orange, red, and white light are designed. Notably, the green- and orange-light-emitting devices achieved an $EQE_{EL}$ exceeding 8.5% and a $PCE_{PV}$ of about 0.5%, whereas MF devices emitting blue and white light exhibited an $EQE_{EL}$ of approximately 2% and a $PCE_{PV}$ exceeding 1% (Fig. 1).

## Results

### Device architecture and design for MF operation

The basic structure of the proposed MF device is illustrated in Fig. 2. This structure consists of a hole injection/extraction layer (HIL), a donor–acceptor pair responsible for exciton generation, and an electron injection/extraction layer (EIL). During EL emission, holes are injected from the anode and electrons from the cathode, leading to exciton formation and subsequent emission at the donor/acceptor interface (Fig. 2b). In this process, the observed emission may originate not only from the CT state exciplex of the donor–acceptor pair but also from the intrinsic emission of either the donor or the acceptor. Conversely, the charge and exciton dynamics during solar energy conversion are depicted in Fig. 2c, using the case where the donor absorbs light to generate electricity. Upon solar irradiation, the donor enters an excited state. The exciton's electron then transfers to the acceptor, forming a CT state. The electron is subsequently extracted at the cathode, whereas the hole is extracted at the anode, generating electrical power. The $E_{CT}$ of this exciton is lower than the energy difference between the donor's ionisation energy (IE) and the acceptor's EA, denoted as the final charge-separated (CS) state energy ($E_{CS}$), by an amount corresponding to the exciton binding energy ($E_b$). $E_b$ can be determined from the difference between $E_{CS}$, obtained using techniques, such as low-energy inverse photoemission spectroscopy (LEIPS), and $E_{CT}$, derived from the EL spectrum of the exciplex[29,36]. By precisely controlling excitons in this manner, we can possibly realise an MF device that integrates both EL and PV functions.

For the realisation of a highly efficient MF device, it is essential to select a donor–acceptor pair with an energy gap ($E_{CS}$) exceeding 2 eV to ensure a large $E_{CT}$. In conventional OPVs, donor–acceptor pairs with $E_{CT}$ below 1.5 eV have commonly been employed, as this design strategy enables the absorption of a broad solar spectrum, thereby achieving high power conversion efficiency[3–5]. However, in MF devices with $E_{CT}$ below 1.5 eV, the efficiency of TTA-UC is low, making it difficult to achieve high $EQE_{EL}$. Therefore, to realise efficient visible-light emission, it is necessary to increase $E_{CS}$ and generate excitons with $E_{CT}$ exceeding 2 eV[22,23,29]. To meet this requirement, we employed the donor and acceptor materials widely used in OLEDs, as shown in Fig. 2d (Supplementary Table 1). In particular, MR-TADF materials exhibit not only high emission efficiency but also small Stokes shift, resulting in

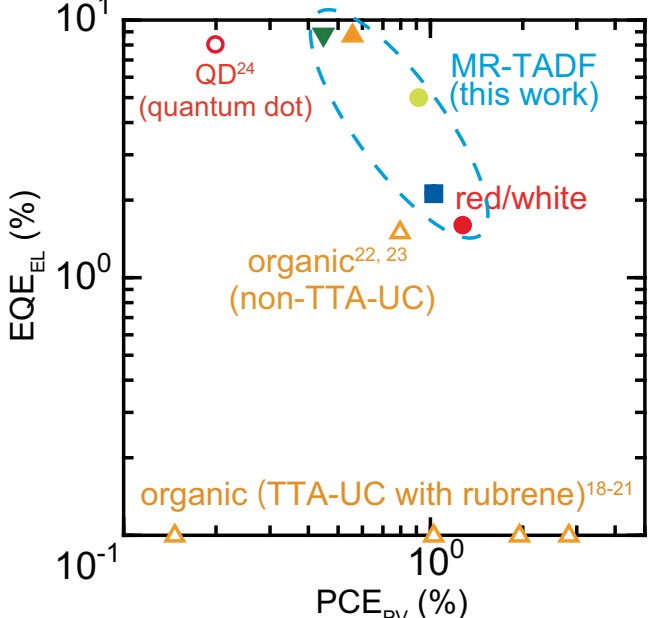

**Fig. 1 | Comparison of MF devices in terms of $PCE_{PV}$ and $EQE_{EL}$.** Open symbols indicate data from previous reports, whereas solid symbols represent data obtained in this work. The colour of each symbol corresponds to the OLED emission colour. $EQE_{EL}$ of all previously reported devices using rubrene is below 0.001%.

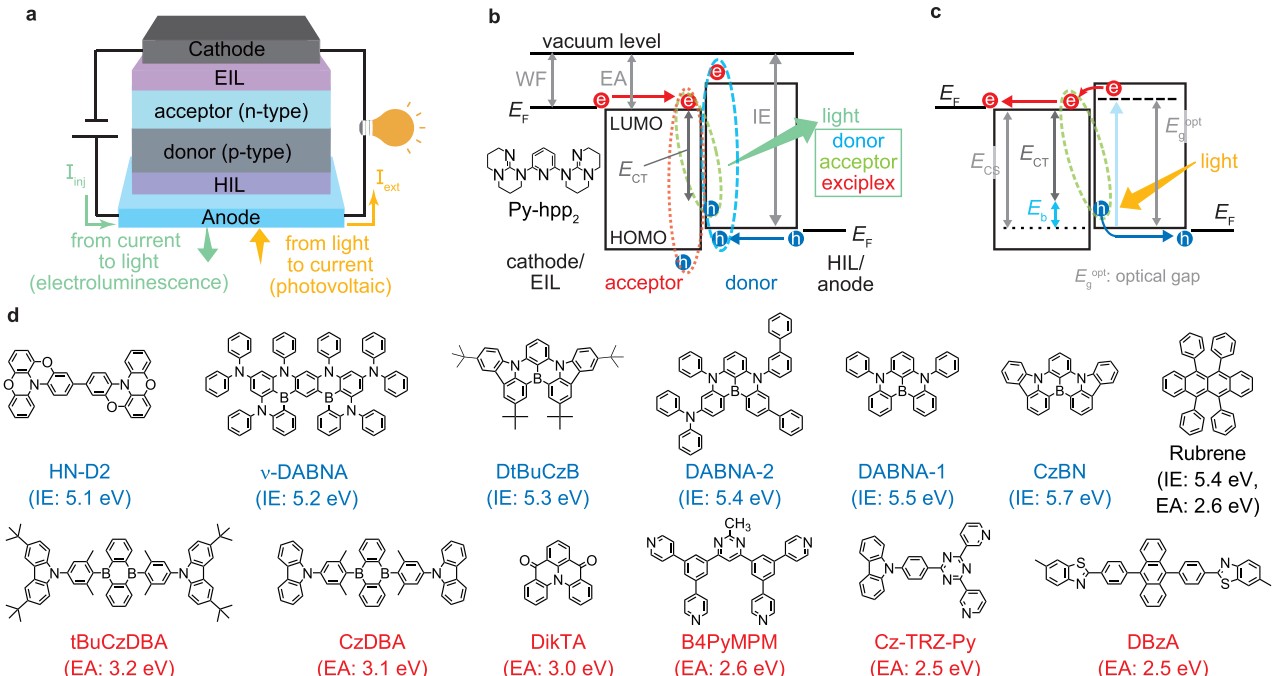

**Fig. 2 | Management of carriers and energies for efficient MF devices.**
**a** Schematic of MF devices with directions of charge flow for electroluminescence and photovoltaic. **b,c** Energy band diagram and the carrier/exciton dynamics in MF devices. **d** Chemical structure of materials used as the donor (blue characters) and acceptor (red characters). Rubrene is used as both the donor and the acceptor.

strong absorption in the solar spectrum[37–39]. Moreover, previous studies have shown that suppressing exciton losses due to triplet excited states is crucial for improving $PCE_{PV}$[15,40,41]. From the perspective of triplet energy ($E_T$) confinement, MR-TADF materials, which exhibit a small energy gap between the singlet and triplet states, are well suited for achieving both high $EQE_{EL}$ and $PCE_{PV}$. Therefore, the utilisation of MR-TADF materials is expected to enable the development of high-performance MF devices. As acceptor materials, we selected TADF materials with an EA exceeding 2 eV, such as 5,10-bis(4-(9H-carbazol-9-yl)−2,6-dimethylphenyl)−5,10-dihydroboranthrene (CzDBA) and 13b-aza-naphtho(3,2,1-de)anthracene-5,9-dione (DikTA), in addition to electron transport materials (ETMs) with relatively high $E_T$, such as 4,6-bis(3,5-di(pyridin-4-yl)phenyl)−2-methylpyrimidin (B4PyMPM) and 9-(4-(4,6-dipyridin −1,3,5-triazin-2-yl)phenyl)−9H-carbazole (Cz-TRZ-Py)[42–45]. The use of Py-hpp₂ enables the reduction in the work function of the cathode to approximately 2.1 eV, allowing for efficient charge collection even from materials with an EA of 3 eV or lower[45]. Conversely, 3,3'-bi[1,4]benzoxazino[2,3,4-kl]phenoxazine (HN-D2), rubrene, and 9,10-bis[4-(6-methylbenzothiazol-2-yl)phenyl]anthracene (DBzA), which exhibit a relatively low $E_T$, were used as reference materials[46]. In this study, we systematically adjusted the donor–acceptor combinations shown in Fig. 2d to achieve $E_{CS}$ values in the range of 2.0–2.7 eV and evaluated the corresponding device performance (Supplementary Table 2).

## Dependence of EL on donor properties

Figure 3 presents the characteristics of MF devices fabricated by varying the donor material while fixing the acceptor material as CzDBA. The EL characteristics shown in Fig. 3a indicate that all devices exhibit current flow at applied voltages below 2 V, reaching a current density of approximately 100 mA cm⁻² at around 4 V. This observation suggests an efficient charge injection at the electrode/organic semiconductor interface, while the low onset voltage for emission can also be attributed to the small $E_b$ of the CT state, consistent with previous reports[23]. As discussed later, many of the devices investigated in this study indeed involve donor–acceptor pairs with relatively small CT-

state $E_b$, which further supports the observed low-voltage EL operation. As shown in Fig. 3b, c, both luminance and $EQE_{EL}$ strongly depend on the donor material, with a maximum $EQE_{EL}$ of approximately 1.2%. Notably, only the devices fabricated employing donors with high $E_T$ values exhibited $EQE_{EL}$ values exceeding 0.5% (Supplementary Fig. 1a). In contrast, the low $EQE_{EL}$ observed in the device with rubrene is likely attributed to the low efficiency of TTA-UC. Whereas the device with rubrene exhibited emission from rubrene facilitated by TTA, all other devices predominantly exhibited EL spectra with peaks above 600 nm (Fig. 3d). Given that CzDBA has an emission peak of around 550 nm, the observed EL spectra above 600 nm are attributed to emission from the CT state (Supplementary Fig. 2a)[8–13]. From the onset of the EL spectra, the $E_{CT}$ of MF devices, excluding the device with rubrene, was estimated to be in the range of 1.96–2.5 eV (Supplementary Table 2)[29]. Furthermore, differences in donor emission intensity within the 450–550 nm wavelength range provide insights into the effect of $E_b$ on exciton and charge behaviour (Supplementary Fig. 3a). As shown in Fig. 3e, a lower $E_b$ correlates with a higher donor emission intensity relative to exciplex EL emission. This trend suggests that at the donor/acceptor interface, a low $E_b$ facilitates exciton dissociation, enabling electrons to migrate to the donor side and contribute to emission (inset in Fig. 3e). A comparison between devices fabricated utilising $N^7,N^7,N^{13},N^{13},5,9,11,15$-octaphenyl-5,9,11,15-tetrahydro-5,9,11,15-tetraaza-19b,20b-diboradinaphtho[3,2,1-de:1',2',3'-jk]pentacene-7,13-diamine (ν-DABNA) and 9-([1,1'-biphenyl]−3-yl)-N,N,5,11-tetraphenyl-5,9-dihydro-5,9-diaza-13b-boranaphtho [3,2,1-de]anthracen-3-amine (DABNA-2), both with an EA of 1.4 eV, revealed that the device incorporating ν-DABNA, which has a lower $E_b$, exhibited a higher emission intensity. This result is notable given that the photoluminescent quantum yield (PLQY) of the neat film of ν-DABNA ( ˜12%) is lower than that of DABNA-2 ( ˜44%), yet a higher emission intensity was observed (Supplementary Table 3). Although this emission intensity is likely affected by the difference in EA between the donor and the acceptor, similar results were obtained even when donors with an EA of 2.1 eV were used (Supplementary Figs. 3a and 3b). These findings suggest that a lower $E_b$ facilitates charge dissociation within the CT state, leading to an

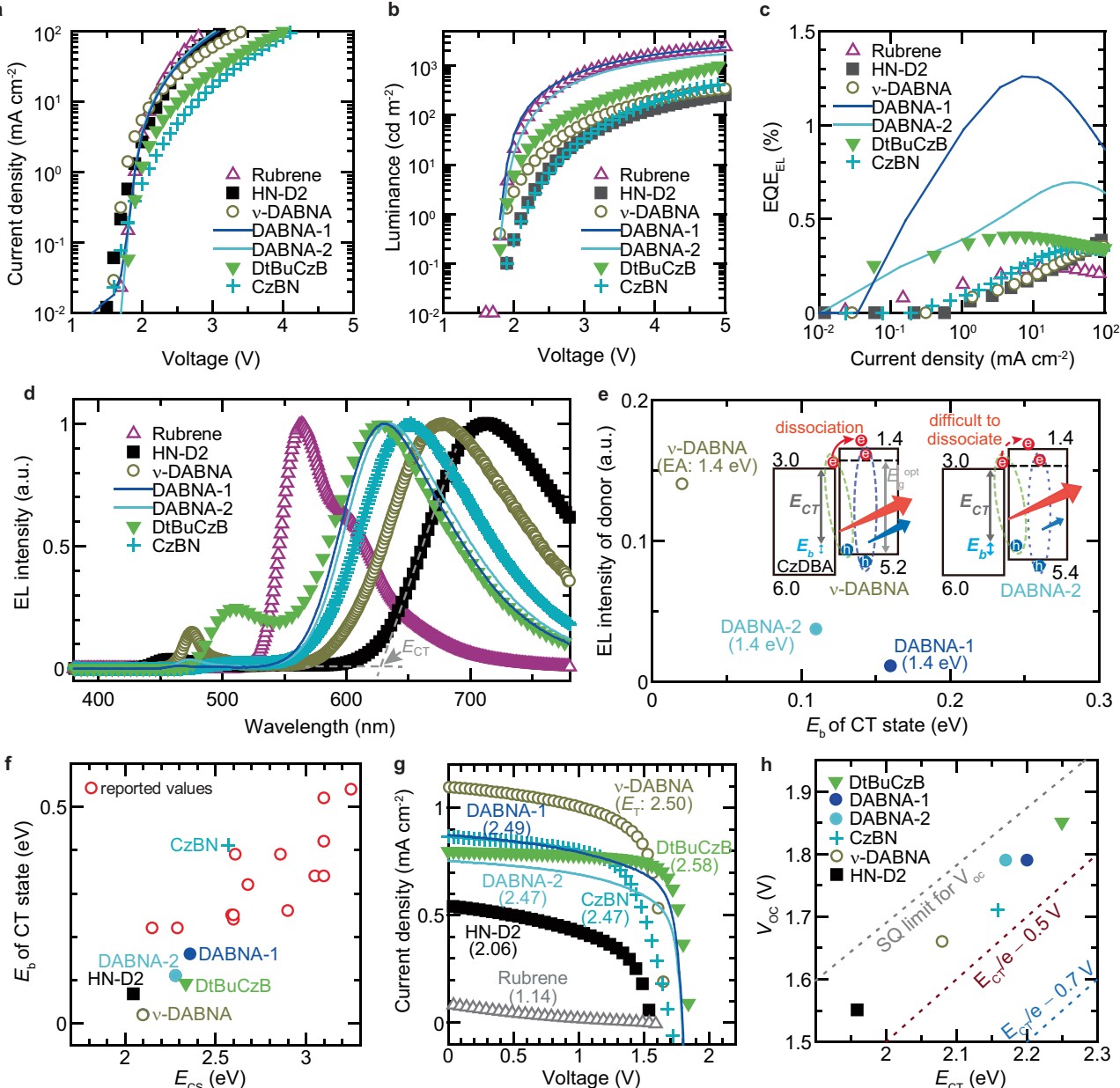

**Fig. 3 | Characteristics of MF devices fabricated using CzDBA as the acceptor.**
**a,b** Current density–voltage (**a**) and luminance–voltage (**b**) characteristics of devices with different donors. (**c**), EQE$_{EL}$ traces of devices for electroluminescence. **d** Electroluminescence spectra of devices. (**e**), Correlation between EL intensity of donor and $E_b$ of CT state. Inset: Schematic of emission mechanism from both the exciplex and the donor. (**f**), $E_b$ of CT state as function of $E_{CS}$. (**g**) Current–density voltage curves of devices measured under simulated AM1.5 G solar illumination. **h** Open-circuit voltage as a function of $E_{CT}$.

increase in donor emission intensity. In this study, several donor/ acceptor interfaces exhibited an $E_b$ of less than 0.2 eV (Fig. 3f). Although $E_b$ increases with $E_{CS}$, many $E_b$ values are smaller than those observed in previous studies using typical arylamine-based donors and triazine-based acceptors[29]. Our results demonstrate that appropriate donor–acceptor combinations can reduce the $E_b$ of the CT state to below 0.2 eV. It has been reported that non-fullerene acceptors exhibit a lower $E_b$ when the number of fused rings and steric hindrance are greater[31]. Since MR-TADF materials and HN-D2 have a greater number of fused rings than typical arylamine-based donors, it is considered that the $E_b$ of the CT state was reduced. Although it is difficult to quantitatively discuss the effect of the number of fused rings on $E_b$ because the MR-TADF donors used in this study possess different fused-ring geometries, a clear influence of steric hindrance has been observed. A comparison between indolo[3,2,1-de]indolo[3′,2′,1′:8,1]

[1,4]benzazaborino[2,3,4-kl]phenazaborine (CzBN) and 2,6-bis(3,6-di-tert-butyl-9H-carbazol-9-yl)boron (DtBuCzB) suggests that the steric hindrance of the tertial butyl group is also effective in reducing $E_b$. Furthermore, a noticeable difference in $E_b$ was also identified between 5,9-Diphenyl-5,9-diaza-13b-boranaphtho[3,2,1-de]anthracene (DABNA-1) and DABNA-2, which likely originates from the different degrees of steric hindrance around their boron centres. A more detailed discussion on the impact of $E_b$ on emission characteristics will be presented in a later section.

## Photovoltaic performance and energy-level correlations

Figure 3g presents the current density–voltage ($J–V$) characteristics of the devices measured under simulated AM1.5 G solar illumination. Owing to the large energy level difference between $E_{CS}$ and $E_{CT}$, most devices exhibited a high $V_{OC}$ exceeding 1.5 V (Supplementary Table 2).

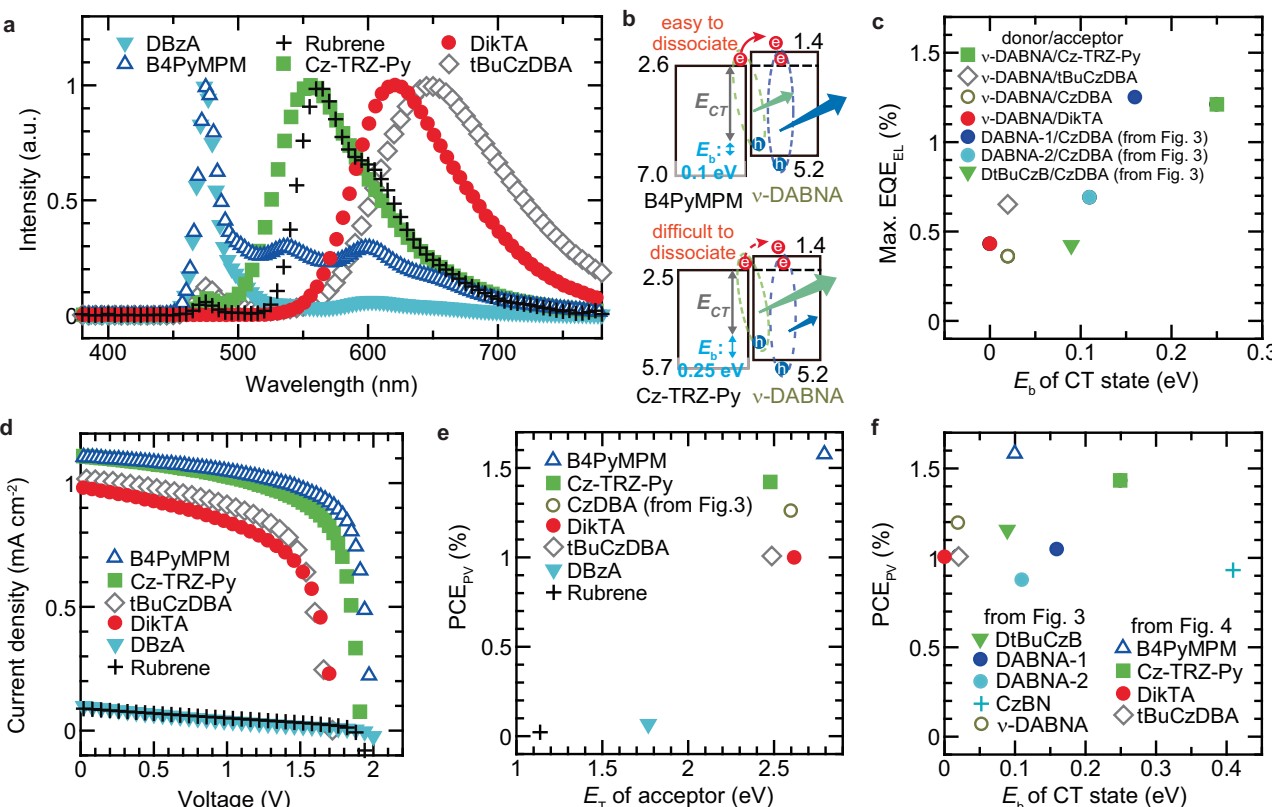

**Fig. 4 | Characteristics of MF devices fabricated using v-DABNA as the donor.**
**a** Electroluminescence spectra of devices. **b** Schematic of mechanism of emission from both the exciplex and the donor. **c** Correlation between maximum $EQE_{EL}$ and $E_b$ of CT state. **d** Current–density voltage curves of devices measured under simulated AM1.5 G solar illumination. **e** $PCE_{PV}$ as a function of $E_T$ of acceptor. **f** $PCE_{PV}$ of MF devices fabricated using MR-TADF material as a function of $E_b$ of CT state.

In recent studies, devices with similarly high $V_{OC}$ have been demonstrated; however, their short-circuit current density ($J_{SC}$) was limited to approximately 0.6 mA/cm² and no devices with $PCE_{PV}$s exceeding 1% have been demonstrated[22,23]. In contrast, four devices with an MR-TADF material as the donor achieved a $PCE_{PV}$ of approximately 1% (Supplementary Table 2). $PCE_{PV}$ exhibits a strong correlation with the $E_T$ of the donor (Supplementary Fig. 4). This high $PCE_{PV}$ is attributed to the strong absorption of the MR-TADF material and its high $E_T$, which effectively confines exciton energy. The photovoltaic EQE of devices employing MR-TADF materials as donors reached approximately 25%, which is comparable to the values reported for state-of-the-art OPDs based on MR-TADF materials (Supplementary Fig. 5)[34]. For comparison, we fabricated an MF device with 3Cz–Ph–TRZ as the donor, a representative donor–acceptor-type TADF material with an IE of 5.2 eV (Supplementary Fig. 6)[47]. However, this device exhibited a low $PCE_{PV}$, primarily because of the poor absorption of 3Cz–Ph–TRZ in the solar spectrum. Additionally, the presence of CzDBA emission in the EL spectrum suggests that CT state formation at the donor/acceptor interface was suppressed. Although HN-D2 exhibits absorption at shorter wavelengths than MR-TADF materials, rubrene absorbs over a broader wavelength range (Supplementary Fig. 2b). Nevertheless, the $J_{SC}$, fill factor (FF), and $PCE_{PV}$ of the rubrene-based device were significantly lower than those of the other devices. Rubrene has been reported to achieve $V_{OC}$ = 0.9 V, $J_{SC}$ > 2 mA/cm², and $PCE_{PV}$ > 2% in stacked OPVs with $C_{60}$[18]. However, in our rubrene–CzDBA-based MF device, no exciplex emission was observed, yet a $V_{OC}$ of 1.85 V was obtained. This suggests that $E_{CT}$ exceeds 2 eV, which is significantly higher than that of conventional donor–acceptor systems, such as rubrene–$C_{60}$. The $E_T$ of rubrene is 1.14 eV, meaning that high-$E_{CT}$ excitons formed at the rubrene/CzDBA interface are likely to undergo rapid nonradiative deactivation[15,40,41]. Fig. 3h shows the correlation

between $V_{OC}$ and $E_{CT}$. In recent studies, OPVs with $E_{CT}$ values exceeding 2 eV have been reported to exhibit small $V_{OC}$ losses, and similar results were obtained in this study[23]. For conventional OPVs with lower $E_{CT}$, the $V_{OC}$ loss is typically large, and $V_{OC}$ is reported to fall within the range of $V_{OC} = E_{CT}/e - 0.5 \sim E_{CT}/e - 0.7$ V[23,48]. However, as shown in Fig. 3h, most devices in this study exhibited small $V_{OC}$ losses, which can be attributed to the low $E_b$. These results indicate that the use of MR-TADF materials enables the realisation of MF devices with both high $PCE_{PV}$ and minimal $V_{OC}$ loss. In MF devices with CzDBA as the acceptor, which emits orange light, EL emission was limited to the orange–red spectral range[42]. To explore the feasibility of MF devices with diverse emission colours, we fabricated devices with different donor–acceptor combinations.

## Emission colour tuning and role of exciton binding energy

When v-DABNA was used as the donor with various acceptors, MF devices exhibited a wide range of emission colours (Fig. 4a). As shown in Supplementary Figs. 7a–c, the $J$–$V$–$L$ characteristics and $EQE_{EL}$ exhibited values similar to those of MF devices with CzDBA as the acceptor. Additionally, a significant decrease in $EQE_{EL}$ was observed in devices fabricated using acceptors with a lower $E_T$, such as rubrene and DBzA (Supplementary Fig. 1b). A particularly noteworthy observation is the differences in EL spectra between devices with B4PyMPM and Cz–TRZ–Py as acceptors, despite both having comparable EAs. The device incorporating Cz–TRZ–Py exhibited strong exciplex emission, whereas the device with B4PyMPM showed predominantly v-DABNA emission. This difference can be understood by considering $E_b$, as illustrated in Fig. 4b. In the device with B4PyMPM, which has an $E_b$ of 0.1 eV, excitons readily dissociate, leading to strong donor emission. Conversely, in the device with Cz–TRZ–Py, which has a relatively high $E_b$ of 0.25 eV, the exciplex emission was predominantly observed. The

high $E_b$ observed in the device with Cz–TRZ–Py is considered to be due to the localisation of the LUMO on the triazine side, as well as the absence of steric hindrance (Supplementary Fig. 8)[47]. For all the CT states examined so far, the least-squares fitting of $E_b$ as a function of $E_{CS}$ yielded a slope of approximately 0.3, which is intriguingly similar to the slope of 0.25 reported for isolated molecules[30]. Because reports on the $E_b$ of CT states are still scarce, we anticipate that acquiring more data will allow a more detailed analysis in the future. Figure 4c shows the correlation between the maximum $EQE_{EL}$ and $E_b$ in devices where exciplex emission was the dominant luminescence. The observed trend of lower $EQE_{EL}$ for lower $E_b$ suggests that a certain threshold $E_b$ is required for the formation and radiative recombination of CT states. Furthermore, in donor–acceptor blend films, a positive correlation was observed between $E_b$ and the PLQY of exciplex emission (Supplementary Fig. 9). While the PLQY of the individual donor and acceptor may also influence the exciplex PLQY, the current dataset is limited, and further data collection will be required to clarify these potential effects. These findings highlight the strong effect of $E_b$ on both the emission colour and efficiency of MF devices.

Figure 4d presents the $J$–$V$ curves of devices measured under simulated AM1.5 G solar illumination. As similarly observed in Fig. 3g high $V_{OC}$ exceeding 1.5 V was achieved in many of the devices, with a maximum $PCE_{PV}$ of approximately 1.5%. As shown in Fig. 4e strong correlation was also observed between $PCE_{PV}$ and the acceptor's $E_T$. Since light absorption in the devices shown in Fig. 4 is primarily attributed to v-DABNA, the correlation between $J_{SC}$ and $E_T$ was analysed to further elucidate the effect of $E_T$ on exciton generation (Supplementary Fig. 10). High $J_{SC}$ was obtained for acceptors with $E_T$ above 2.4 eV, indicating that the insufficient confinement of $E_T$ leads to exciton quenching. On the other hand, in the MF devices fabricated employing MR-TDAF materials shown in Figs. 3 and 4, high $PCE_{PV}$ was achieved even when $E_b$ was around 0.4 eV, and no clear correlation between $PCE_{PV}$ and $E_b$ was observed (Fig. 4f). This can be attributed to the simple donor/acceptor stacked structure of the devices. The electron mobility of a tBuCzDBA neat film is on the order of $10^{-6}$ cm$^2$ V$^{-1}$ s$^{-1}$, whereas that of a CzDBA neat film is on the order of $10^{-5}$ cm$^2$ V$^{-1}$ s$^{-1}$. Nevertheless, the $PCE_{PV}$ of v-DABNA-based MF devices was comparable in both cases, suggesting that a one-order difference in electron mobility exerts only a minor influence on $PCE_{PV}$ (Figs. 4e and 4f)[49,50]. In contrast, in the OLED characteristics ($J$–$V$ curves) of the v -DABNA-based MF devices, those employing tBuCzDBA required a higher operating voltage (Fig. 3a and Supplementary Fig. 7a). Although no clear correlation was observed between $E_b$ and $PCE_{PV}$ (Fig. 4f), a trend was identified in which a larger $E_{CT}$ led to a higher $PCE_{PV}$ (Supplementary Fig. 11a). This can be attributed to the fact that an increased $E_{CT}$ results in a higher $V_{OC}$ (Supplementary Fig. 11b). These $PCE_{PV}$ values were higher than those reported for recently developed MF devices with $V_{OC}$ exceeding 1.5 V, demonstrating the effectiveness of MR-TADF materials[23].

## Realization of high-efficiency multicolour MF devices

On the basis of the obtained EL and PV characteristics and physical properties, such as $E_b$, MF devices exhibiting various emission colours were successfully realised. The device configurations, along with their emission and PV characteristics, are shown in Fig. 5 (Supplementary Fig. 12, Supplementary Table 4). For blue-, green-, and yellow-light-emitting MF devices, DABNA-2 was selected as the donor because of its high PLQY in both neat and blended films (Supplementary Table 3). For demonstrating the blue-light-emitting device, B4PyMPM was chosen as the acceptor to minimise $E_b$, ensuring that the emission originated from DABNA-2. Although the $E_b$ of the CT state in the blue-light-emitting device was estimated to be 0.21 eV, blue emission from DABNA-2 was mainly observed. The blue-light-emitting device exhibits an $EQE_{EL}$ of 2% and a $PCE_{PV}$ of 1%. To utilise the high PLQY of the DABNA-2 doped with B4PyMPM film (44%), a doped layer was

introduced into the green-light-emitting device. As a result, an $EQE_{EL}$ exceeding 8.5% was achieved. Considering the PLQY of 44% and a light outcoupling efficiency of approximately 20%, this $EQE_{EL}$ suggests that almost all generated excitons were efficiently converted into light[51]. However, the $PCE_{PV}$ was lower than that of a simple donor/acceptor stacked device. This reduction in $PCE_{PV}$ is attributed to the large energy differences in IE and EA between DABNA-2 and B4PyMPM, which led to charge trapping. Furthermore, by using Cz-TRZ-Py, which is an acceptor with higher $E_b$, in the same device structure as the blue device, we obtained a yellow-light-emitting MF device. The $E_b$ of the CT state of DABNA–2/Cz–TRZ–Py was estimated to be 0.41 eV. This device exhibited a similar $PCE_{PV}$ (~1%) to the blue-light-emitting device and achieved an $EQE_{EL}$ of approximately 5%. To further examine the possible influence of carrier injection and transport on the EL spectra, we compared four MF devices using v-DABNA or DABNA-2 as donors and B4PyMPM or Cz-TRZ-Py as acceptors (Supplementary Fig. 13). The results suggest that differences in hole-injection and transport between v-DABNA and DABNA-2 may have some influence on exciton dissociation, but the effect appears less significant than that of $E_b$. In the devices employing v-DABNA as the donor, the smaller IE of v-DABNA led to higher current density and earlier hole arrival at the donor/acceptor interface. In such a situation, increased hole flux to the interface would generally be expected to suppress donor emission; however, strong donor emission was observed specifically in the devices with B4PyMPM, which yielded smaller $E_b$ values. These results indicate that, for emission-colour modulation in MF devices, the influence of $E_b$ outweighs that of donor-dependent carrier injection and transport. By blending tBuCzDBA and DtBuCzB, we realised an orange-light-emitting MF device, demonstrating comparable $PCE_{PV}$ and $EQE_{EL}$ to the green-light-emitting device. To further clarify the emission origin, the PL spectra of the three mixed films corresponding to the green-, yellow-, and orange-emitting devices are shown in Supplementary Fig. 14. These results confirm that the green, yellow, and orange EL emissions mainly originate from CT states. MF devices, such as the green- and orange-light-emitting devices, which simultaneously achieved an $EQE_{EL}$ exceeding 8.5% and a $PCE_{PV}$ of about 0.5%, were not reported previously, marking the realisation of the highest-performance MF devices to date (Fig. 1, Supplementary Tables 4, 5 and 6). These devices employed doped layers that exhibited higher PLQY than the corresponding undoped films, and the expansion of the recombination region achieved through the introduction of the doped layer further enhanced $EQE_{EL}$. This combination of improved radiative efficiency and a broadened recombination region likely explains the higher $EQE_{EL}$ observed in the green and orange devices compared with the blue and red ones. By precisely controlling the behaviour of charges, excitons, and CT states, we successfully realised MF devices using organic semiconductors with both high EL and power conversion efficiencies, which was previously considered challenging. It was also discovered that by doping additional emissive materials into the device, in addition to the donor–acceptor materials required for PV conversion, the emission colour could be further expanded (R/W device). The MF device doped with tris(1-phenylisoquinoline)iridium(III) [Ir(piq)$_3$], a red phosphorescent dopant, exhibited red emission at low current densities and white emission at high current densities[52]. In contrast, other MF devices showed minimal emission colour variation with current density (Supplementary Fig. 15).

The realisation of MF devices exhibiting various emission colours significantly broadens their potential applications. The successful demonstration of MF devices capable of RGB emission suggests that, when implemented in display technology, display panels could be designed to harvest ambient light for charging when not in use. Furthermore, the realisation of white emission highlights the feasibility of emerging applications, such as self-powered lighting. Currently, the operational stability of these MF devices remains limited; however, optimising materials and device architectures is

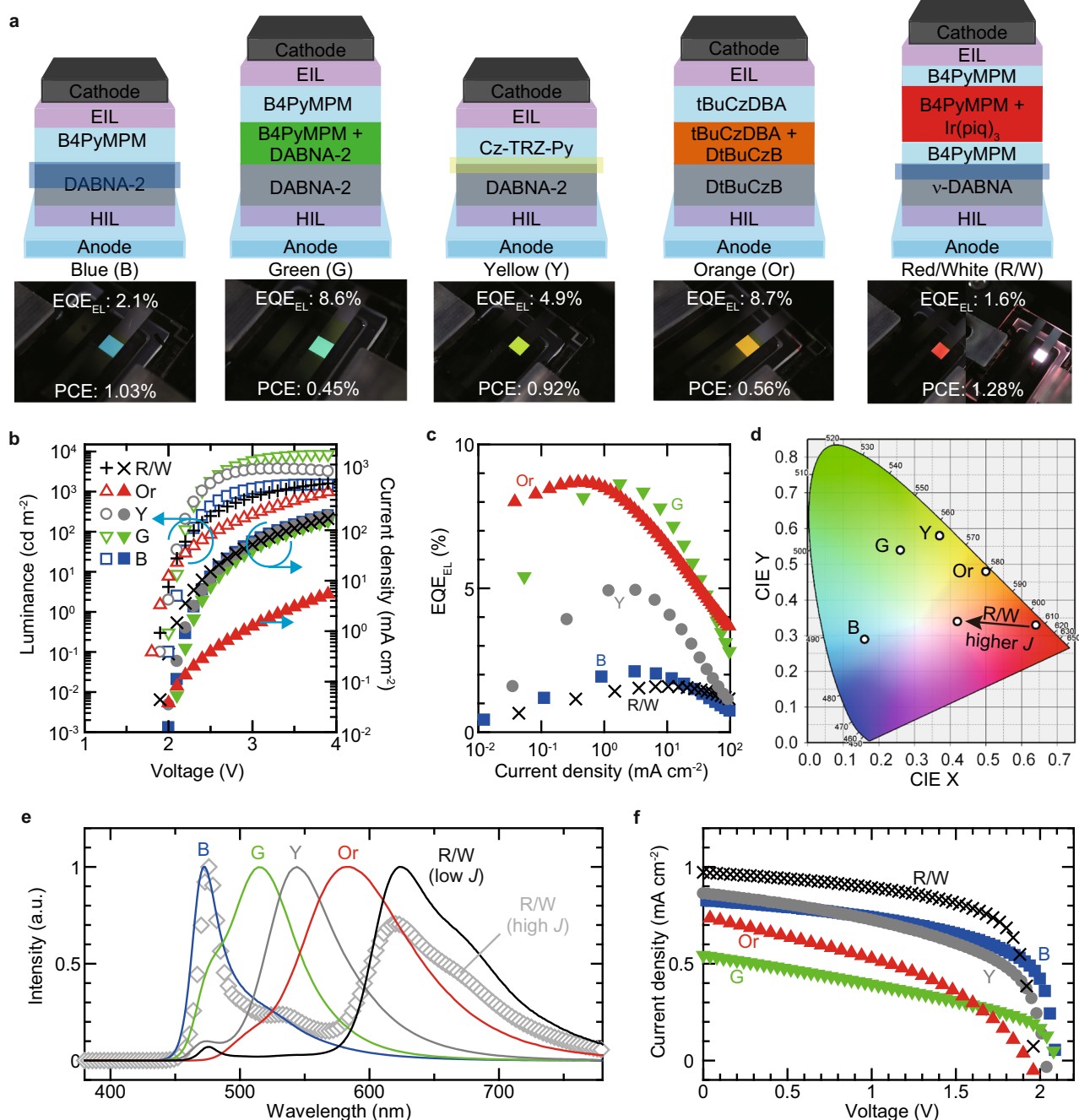

**Fig. 5 | Characteristics of optimised devices with various emission colours.**
**a** Schematic illustration and picture of optimised devices with various emission colours. **b** Luminance–voltage–current density characteristics of devices. **c** $EQE_{EL}$–current density characteristics of the devices. **d** Corresponding 1931 CIE coordinates of device emission. **e** Electroluminescence spectra of devices. **f** Current density–voltage curves of devices measured under simulated AM1.5 G solar illumination.

expected to extend their lifetime (Supplementary Fig. 16). The blue device exhibited a shorter EL lifetime but a longer PV lifetime than the yellow device. Although both devices showed comparable $PCE_{PV}$ values, the higher EL efficiency of the yellow device was accompanied by a shorter PV lifetime, suggesting that achieving both high efficiency and long-term stability remains challenging. Continued material development will be essential to address this trade-off. Notably, in 2024, an OPV, which features a donor/acceptor stacked structure similar to that used in this study, was reported to achieve a $PCE_{PV}$ exceeding 20% with high stability[4]. Building on these findings, further improvements in both the efficiency and stability of MF devices can be anticipated. In particular, the development of MR-

TADF materials with high PLQY in both neat and highly doped films holds great promise for significantly enhancing the $EQE_{EL}$ of MF devices. In this study, although the $PCE_{PV}$ was limited to a maximum of ~1.5%, we obtained several indications that point toward possible strategies for improvement. For example, in MF devices incorporating a mixed film of v-DABNA and CzDBA, increasing the mixed film thickness led to a higher $J_{SC}$ but a reduced FF (Supplementary Fig. 17). This behaviour can be explained by the fact that a thicker mixed layer increases the number of charge-generating interfaces, while the decrease in FF is likely caused by a large offset between the IE and EA of v-DABNA and CzDBA. Furthermore, in MF devices employing HN-D2 together with DikTA, which possesses a complementary

absorption range, we observed an enhancement in $J_{SC}$ similar to that typically seen in bulk-heterojunction OPVs (Supplementary Fig. 18)[43]. These results suggest that the use of mixed layers containing donor/acceptor pairs with complementary absorption spectra, combined with carefully designed energy levels that do not hinder charge transport, can provide a pathway to further improvement in $PCE_{PV}$. To achieve a balance between high $EQE_{EL}$ and $PCE_{PV}$, a deeper understanding of the influence of $E_b$ is also crucial. Although this study mainly focused on simple donor/acceptor-stacked structures, further clarification of how $E_b$ affects device characteristics in doped-film-based MF devices will be essential for identifying its optimal range.

## Discussion

In this study, we demonstrated the simultaneous realisation of both high $EQE_{EL}$ and $PCE_{PV}$ in a single organic MF device. By employing MR-TADF materials with both strong absorption and high photoluminescence efficiency, we successfully overcame efficiency trade-offs and simultaneously enhanced $EQE_{EL}$ and PV performance. The use of MR-TADF materials led to significant reductions in $E_b$ compared with conventional organic semiconductors, with donor–acceptor $E_b$ values ranging from approximately 0 to 0.4 eV. Notably, MF devices with lower $E_b$ exhibited remarkably small $V_{OC}$ losses, whereas its impact on $PCE_{PV}$ remained relatively minor. $E_b$ proved to be a key factor for determining both emission colour and emission efficiency, underscoring its critical role in MF device performance. These findings highlight the importance of precise $E_b$ control for optimising device efficiency and achieving emission colour tuning from blue to red. Ultimately, this work provides a strategy for overcoming efficiency trade-offs in organic devices and paves the way to next-generation applications, such as self-powered displays and lighting, where EL and PV functions are seamlessly integrated into a single platform.

## Methods

### Fabrication of MF devices (MF devices)

The MF devices shown in Figs. 3 and 4 were fabricated on glass substrates coated with a 100-nm-thick ITO layer. Prior to the fabrication of the organic layers, the substrate was cleaned using ultrapurified water and organic solvents, and by UV–ozone treatment. After the UV–ozone treatment, Clevios HIL 1.3 N (supplied by Heraeus Holding GmbH) was spun onto the substrate to form a 10-nm-thick layer. The other organic layers were sequentially deposited onto the substrate. The film structure of the OLEDs was ITO (100 nm)/Clevios HIL 1.3 N (10 nm)/donor (30 nm)/acceptor (30 nm)/Py-hpp$_2$ (3 nm), where Py-hpp$_2$ is 2,6bis(1,3,4,6,7,8-tetrahydro-2H-pyrimido[1,2-a]pyrimido-1-yl)pyridine. After the formation of Py-hpp$_2$, a 100-nm-thick Al layer was deposited as the cathode. The devices were encapsulated using a UV-epoxy resin, a glass cover, and a desiccant in a nitrogen atmosphere after cathode formation.

MF devices with various emission colours (shown in Fig. 5) were fabricated similarly to the simple-structure OLEDs except for the organic layer. The film structure of the MF devices was ITO/Clevios HIL 1.3 N (10 nm)/organic layers/Py-hpp$_2$ (3 nm). The organic layers used for each device were DABNA-2 (30 nm)/B4PyMPM (30 nm) for blue emission, DABNA-2 (30 nm)/DABNA-2 doped with B4PyMPM (50 wt% 10 nm)/B4PyMPM (30 nm) for green emission, DABNA-2 (30 nm)/Cz-TRZ-Py (40 nm) for yellow emission, DtBuCzB (30 nm)/ DtBuCzB doped with tBuCzDBA (50 wt% 20 nm)/ tBuCzDBA (40 nm) for orange emission, and v-DABNA (40 nm)/B4PyMPM (5 nm)/B4PyMPM: Ir(piq)$_3$ (1 wt%, 3 nm)/ B4PyMPM (22 nm) for red/white emission. After the formation of Py-hpp$_2$, a 100-nm-thick Al layer was deposited as the cathode. The devices were encapsulated using a UV-epoxy resin, a glass cover, and a desiccant in a nitrogen atmosphere after cathode formation.

Materials were purchased from Luminescence Technology Corporation and Tokyo Chemical Industry Co., Ltd. We used them after sublimation.

### Device characterization

The EL spectra and luminance were measured using a spectroradiometer (Minolta CS-1000). A digital SourceMeter (Keithley 2400) and a desktop computer were used to operate the devices. We assumed that the emission from MF devices was isotropic so that the luminance was Lambertian; thus, we calculated $EQE_{EL}$ from the luminance, current density, and EL spectra. Current–voltage characteristics in the dark and under solar illumination were measured with a Keithley 2400 at room temperature under ambient conditions. The cells were illuminated with a spectrally mismatch-corrected intensity of 100 mW cm$^{-2}$ (AM1.5 G) provided by a sun simulator (OTENTO-SUNIIIFNS, Bunkokeiki Co., Ltd.), with intensity calibration via a standard silicon photodiode. The device areas were 3 × 3 mm². To confirm the validity of the experimental results, devices with identical structures were fabricated on different experimental days. The variation in both $EQE_{EL}$ and $PCE_{PV}$ among these devices was found to be less than 2%, indicating that the device-to-device variation is small (see Supplementary Fig. 19). Photovoltaic EQE and response spectra of the devices were collected using a Spectral sensitivity measurement system VC-250 (Bunkokeiki Co., Ltd.) under zero-bias conditions.

### IE and EA measurements

The IE of the materials was determined by the spectroscopic measurement of photoemission in the air (AC-3, Rikenkeiki). The LEIPS spectra of organic thin films deposited on glass/ITO using a vacuum evaporation system were measured using an LEIPS measurement system (ALS Technology Co., Ltd.). No discernible dependence on the sample current or photon energy was observed, confirming that the LEIPS spectra were free from sample charging. A bandpass filter with a centre wavelength of 260 nm was used. The onset of an LEIPS spectrum was determined as the intersection of the straight line fitted to the onset region of the spectrum and the baseline.

### Photoluminescence (PL) measurement

The 50-nm-thick organic films used for optical measurements were fabricated on clean quartz substrates by thermal evaporation. The PL spectra of the films and the transient PL characteristics were recorded using a spectrofluorometer (Horiba Jobin Yvon, FluoroMax-4). The excitation wavelength for all PL measurements was 350 or 355 nm. The sample PLQY was measured using a PL quantum yield measurement system (Hamamatsu Photonics, Quantaurus-QY).

## Data availability

All data supporting the findings of this study are provided in the article and its Supplementary Information. No additional raw data files are required to interpret or reproduce the results.

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

## Acknowledgements

The authors thank Heraeus Holding GmbH for supplying Clevios HIL 1.3. T.H. acknowledges Japan Science and Technology Agency (JST) CREST (grant no. JPMJCR22B3, T.H.). This research was financially supported by the Japan Society for the Promotion of Science (JSPS) KAKENHI Grant-in-Aid for Scientific Research (grant no. 24K23071, H.F.).

## Author contributions

T.O. and Y.A. fabricated the MF devices and measured their characteristics with help from T.Ok., T.S., and H.S. T.H. and T.Sh. revised the manuscript. H.F. supervised the project and wrote the manuscript. All authors discussed the results and contributed to the paper.

## Competing interests

The authors declare no competing interests.
