## [Transparent Peer Review file · Nature Communications]

A pathway to coexistence of electroluminescence and photovoltaic conversion in organic devices

Corresponding Author: Professor Hirohiko Fukagawa

Version 0:

Reviewer comments:

Reviewer #1

(Remarks to the Author)

This work tackles the persistent trade-off between electroluminescence and photovoltaic efficiency, which has long hindered progress in the field of organic optoelectronics. By using MR-TADF materials and tuning donor–acceptor combinations to control the CT state and its exciton binding energy, the authors demonstrate multifunctional organic devices with emission ranging from blue to red, and with moderate PCE (up to ~1.5%) and EQEEL (>8.5%). Although the work is novel and potentially impactful, several issues in device characterization and mechanistic interpretation should be addressed to meet the standards of Nature Communications.

1. There is no mention of error bars, device-to-device variation, or number of tested devices. Given the relatively low current densities and PCEs, even small variation can change the overall interpretation. Please provide statistical analysis for EQEEL and PCE.
2. Although the authors report up to PCE of ~1.5%, most devices remain below 1%. Please provide additional discussion or simulations on why PCE is limited despite high EQEEL, and suggest how efficiency could be further improved in future designs.
3. The manuscript asserts that emission is either from donor or CT state. However, there are no clear spectroscopic fingerprints shown to confirm the nature of emission.
4. The device emitting red/white light shows current-dependent emission switching. However, no detailed color rendering metrics (e.g., CRI, CCT) are provided, and the spectral stability under long-term operation is not discussed.
5. In methods, the authors should include more details in the Methods section regarding device encapsulation and measurement conditions.
6. To improve the clarity and conciseness of the main text, I recommend moving some of the redundant or less critical figures to the Supporting Information. This would help streamline the presentation and allow the key results to stand out more clearly.

Reviewer #2

(Remarks to the Author)

In this manuscript, the authors report the simultaneous achievement of relatively high electroluminescence (EL) efficiency and power conversion efficiency (PCE) in a single organic device by employing MR-TADF materials. The authors demonstrated that precise modulation of E_b can regulate both the efficiency and emission color of the devices. This work is of potential interest to the OLED and OPV communities, as it expands the functionality of organic semiconductors toward self-powered displays and lighting applications. I would recommend acceptance of this manuscript after following issues are addressed.

1. Donor- or acceptor-isolated EL spectra should be provided to distinguish emissions originating from CT states or TTA-UC. It is also recommended to include the absorption and PL spectra of the relevant donor and acceptor, eg. CzBDA. Please include the PL spectra of the five devices, particularly for the mixed films. In addition, provide L_{max} , PE_{max} , and CE_{max} for the devices.
2. The authors attributed the variation in donor emission intensity observed when employing different MR-TADF emitters (V-DABNA and DABNA-2) as donors to exciton dissociation induced by a low E_b , while it should be noted that differences in the carrier mobilities and HOMO energy levels of the donors can lead to variations in carrier transport within the devices, thereby affecting the location of the recombination zone, which implies that the shift of the recombination zone might be

responsible for the change in donor EL emission intensity. Therefore, it is necessary to exclude the influence of donor-induced differences in carrier transport in order to further substantiate that the change in E_b is the factor responsible for the variation in donor EL emission intensity.

3. The authors discussed in detail the interfacial CT-state emission of organic MF devices in the manuscript. The modulation of CT-state emission is an effective strategy for improving EL devices, some related works (Adv. Mater. 2016, 28, 239–244; Adv. Mater. 2016, 28, 6758–6765; Adv. Mater. Interfaces 2018, 5, 1800025; Adv. Mater. 2024, 36, 2403584) could be included as references.

4. Although the authors achieved both high EL EQE and PCE in a single MF organic device, the performance of MF devices with emission colors ranging from blue to red, and even white, still exhibits a trend in which PCE decreases as EL EQE increases. How the authors view this issue, along with a brief discussion and outlook, may help guide further improvements in the performance of MF devices.

5. In Fig. S2b, please provide the PLQY for the two materials. This will allow assessment of the effect of E_b on the donor emission intensity when controlling the variable of EA. In the statistical analysis of E_b and E_{cs} in Fig. S6, please clarify the function used. In Fig. S7, is E_b positively correlated with PLQY? Are there other influencing factors? Please elaborate.

6. For Fig. 5, it is suggested to supplement with an energy-level diagram of the device, including E_b , ET, and ECS for both the donor and acceptor, to enable more intuitive comparative analysis for the reader.

7. How does the interfacial morphology of the films affect device performance? How does the carrier mobility of different donor and acceptor materials influence the performance? In the discussion of results, it is recommended to summarize the effects of donor ET, exciplex ECT, ECS, E_b , and PLQY on device EQE and PCE, in order to guide material selection and performance trade-offs in practical applications.

8. The caption “Figure 3f shows the correlation between the maximum EQEEL and E_b ” should be corrected to “Figure 4f shows the correlation between the maximum EQEEL and E_b .”

Reviewer #3

(Remarks to the Author)

This manuscript reports on excellent and original work, discovering molecular donor:acceptor material systems with combining a high EL quantum efficiency and high photovoltaic efficiency. The work is important as it provides important insights into materials design rules allowing to combine a high free charge carrier generation yield upon illumination with a high LED quantum efficiency. Such materials are key to achieve power conversion efficiencies close to the fundamental Shockley-Queisser limit. This work should be published and I only have a few comments which I hope the authors will take into account:

First sentence of the abstract: “Achieving both high electroluminescence (EL) efficiency and power conversion efficiency (PCE) in a single organic device has long been considered challenging, as these processes are fundamentally opposite.” I would argue that these processes are not fundamentally opposite: In fact, the Shockley-Queisser limit for photovoltaics dictates that the highest PCE will be achieved for the absorber with the highest EL efficiency (since that one will have the highest V_{oc}). I would ask the authors to reconsider or soften this statement. I would agree that it is difficult to achieve both high EL and high PV PCE as it involves reducing the binding energy of the interfacial states. But this is not fundamentally impossible.

Also from the abstract “In this study, we present a novel strategy for material selection and device design to overcome this limitation and realise the unprecedented coexistence of high EL and photovoltaic (PV) efficiencies.” Authors should mention already in the abstract that this new strategy is to make use of multiple-resonance thermally activated delayed fluorescence (MR-TADF) materials, which exhibit high emission efficiency and strong absorption.

To avoid confusion, throughout the manuscript, authors should clarify when they mention PCE, that it is photovoltaic PCE (PCE_{PV}) and not LED PCE (electron to photon power conversion efficiency). When they mention EQE, they should clarify when it is LED EQE (electron-to-photon) or photovoltaic EQE (photon-to-electron).

Introduction: “Furthermore, if both EL and PV functions can be combined, it would be possible to realise a mobile display that can be charged using indoor lighting or sunlight when not in use. This would free people from the hassle of charging and significantly reduce the overall power consumption of displays in society.” This statement is partially true, as for the best performance, the PV device should of course absorb part the NIR of the sunlight, which is not possible if it also has to emit visible light.

Introduction: “Thus, in organic-semiconductor-based devices, it is inherently challenging to achieve high performance in both EL and PV functions, which are fundamentally opposite processes.” See my previous comment: if non-radiative pathways are minimized, the photovoltage losses and EQE_{EL} will be simultaneously maximized. They are thus not fundamentally opposite. Decreasing the binding energy might increase the change for the charge carriers to find non-radiative pathways, but it is not a fundamental limitation if non-radiative pathways are minimized.

Results and discussion: “The EL characteristics shown in Fig. 3a indicate that all devices exhibit current flow at applied voltages below 2 V, reaching a current density of approximately 100 mA/cm² at around 4 V. This observation suggests an efficient charge injection at the electrode/organic semiconductor interface.” The low onset voltage for emission is the result of a low excited state binding energy, see reference 19. Even if charge injection and the electrode/organic interface would be efficient, it is still necessary to reduce the CT state binding energy in order to achieve such low onset voltages.

Absorption spectra, or more ideally, photovoltaic external quantum efficiency spectra, are missing from the main text. They would corroborate the statement that MR-TADF materials are strong absorbers.

Version 1:

Reviewer comments:

Reviewer #1

(Remarks to the Author)

The authors have provided detailed and well-structured responses to all reviewers' comments. No major scientific issues remain. The only minor suggestion is that the authors could briefly comment in the Discussion section on the potential stability challenges of MR-TADF materials under simultaneous EL and PV operation, as Supplementary Fig. S16 indicates limited device lifetime.

Reviewer #2

(Remarks to the Author)

The authors have addressed most of the issues. There are still two more questions to be discussed.

1. The authors have demonstrated multifunctional devices with emission colors ranging from blue to red and even white. However, there are significant performance disparities, particularly in EQEEL and PCEPV, among the devices of different colors (e.g., blue/red EQEEL \approx 2% vs. green/orange $>$ 8.5%). It is recommended that the authors provide a deeper analysis of the fundamental reasons behind these differences. Are they related to the intrinsic PLQY of the different MR-TADF materials, variations in charge carrier mobility, or differences in non-radiative decay pathways of the charge-transfer states formed with the respective acceptors?
2. The authors demonstrated that the exciton E_b is a key parameter for tuning both emission color and device efficiency, attributing its reduction to the multi-fused-ring structure and steric hindrance of the MR-TADF materials. It is recommended that the authors further discuss whether this strategy of modulating E_b through molecular structure implies a universal design rule. For instance, can all donors with large, planar, and multi-fused ring structures effectively lower E_b ? Furthermore, is there an optimal range for E_b that best balances high EQEEL and high PCEPV? The current conclusion suggests that " E_b cannot be too low (otherwise CT-state radiative recombination efficiency suffers), but its impact on PV efficiency is relatively minor." Could this conclusion be elaborated more quantitatively?

Reviewer #3

(Remarks to the Author)

authors have taken into account all my comments and have improved the manuscript. I recommend publication of this excellent work.

Our Responses to the Comments of the Reviewers

Reviewer #1

General comment: *This work tackles the persistent trade-off between electroluminescence and photovoltaic efficiency, which has long hindered progress in the field of organic optoelectronics. By using MR-TADF materials and tuning donor–acceptor combinations to control the CT state and its exciton binding energy, the authors demonstrate multifunctional organic devices with emission ranging from blue to red, and with moderate PCE (up to ~1.5%) and EQEEL (>8.5%). Although the work is novel and potentially impactful, several issues in device characterization and mechanistic interpretation should be addressed to meet the standards of Nature Communications.*

Our response: We are grateful to the reviewer for recommending publication of the manuscript in Nature Communications after some revisions. Changes are shown by red-letter in the revised manuscript. (The Supporting Information is written in black letter.)

Comment 1: *There is no mention of error bars, device-to-device variation, or number of tested devices. Given the relatively low current densities and PCEs, even small variation can change the overall interpretation. Please provide statistical analysis for EQEEL and PCE.*

Our response:

We thank the reviewer for raising this important point. To confirm the validity of the experimental results, we fabricated devices with identical structures on different experimental days and compared their performance. The variation in both EQE_{EL} and PCE_{PV} was found to be less than 2%, indicating that the device-to-device variation is small. We have added this information at the end of the *Device characterisation* section, together with a Supplementary figure showing the data. The following discussion has been added in *Device characterisation* section:

To confirm the validity of the experimental results, devices with identical structures were fabricated on different experimental days. The variation in both EQE_{EL} and PCE_{PV} among these devices was found to be less than 2%, indicating that the device-to-device variation is small (see Supplementary Figure 19).

Comment 2: *Although the authors report up to PCE of ~1.5%, most devices remain below 1%. Please provide additional discussion or simulations on why PCE is limited despite high EQEEL, and suggest how efficiency could be further improved in future designs.*

Our response: We thank the reviewer for this constructive comment. We agree that the photovoltaic efficiency of the present MF devices is limited, and we have therefore included additional discussion to address the factors responsible for the relatively low PCE_{PV} and possible strategies for improvement. In particular, we describe how, in MF devices incorporating a mixed film of v-DABNA and CzDBA, increasing the film thickness results in a higher J_{sc} but a reduced FF due to the large offset between the IE and EA of the two materials (Supplementary Fig. 17). We also discuss results from MF devices employing HN-D2 together with DiKTA, which has a complementary absorption range, showing that such a configuration enhances J_{sc} in a manner analogous to conventional bulk-heterojunction OPVs (Supplementary Fig. 18). Based on these results, we suggest that the use of mixed donor/acceptor layers with complementary absorption spectra, together with carefully designed energy levels that facilitate charge transport, provides a pathway to improving PCE_{PV}.

The following discussion has been added before Conclusion section:

In this study, although the PCE_{PV} was limited to a maximum of ~1.5%, we obtained several indications that point toward possible strategies for improvement. For example, in MF devices incorporating a mixed film of v-DABNA and CzDBA, increasing the mixed film thickness led to a

higher J_{SC} but a reduced FF (Supplementary Fig. 17). This behaviour can be explained by the fact that a thicker mixed layer increases the number of charge-generating interfaces, while the decrease in FF is likely caused by a large offset between the IE and EA of v-DABNA and CzDBA. Furthermore, in MF devices employing HN-D2 together with DikTA, which possesses a complementary absorption range, we observed an enhancement in J_{SC} similar to that typically seen in bulk-heterojunction OPVs (Supplementary Fig. 18).⁴³ These results suggest that the use of mixed layers containing donor/acceptor pairs with complementary absorption spectra, combined with carefully designed energy levels that do not hinder charge transport, can provide a pathway to further improvement in PCE_{PV} .

Comment 3: *The manuscript asserts that emission is either from donor or CT state. However, there are no clear spectroscopic fingerprints shown to confirm the nature of emission.*

Our response: We thank the reviewer for this important comment. To clarify the origin of the emission, we have added additional spectroscopic data. Specifically, we included the PL spectrum of a CzDBA neat film in Supplementary Fig. 2a to demonstrate that the EL spectra in Fig. 3 of MF devices other than those using rubrene are indeed CT-state-derived. In addition, because Cz-TRZ-Py is a material originally developed in our group and reported previously, we also added its neat-film PL spectrum in Supplementary Fig. 2a. For the other donor/acceptor materials, the corresponding neat-film PL spectra are available in the cited references. With these additions, the emission from the donor and the CT state can be more clearly distinguished across the presented EL spectra, thereby supporting our assignment of the emission origin in this study.

The manuscript has been changed as follows (pages 8-9):

Original	Given that CzDBA has an emission peak of around 550 nm, the observed EL spectra above 600 nm are attributed to emission from the CT state. ⁸⁻¹³
Corrected	Given that CzDBA has an emission peak of around 550 nm, the observed EL spectra above 600 nm are attributed to emission from the CT state (Supplementary Fig. 2a). ⁸⁻¹³

Comment 4: *The device emitting red/white light shows current-dependent emission switching. However, no detailed color rendering metrics (e.g., CRI, CCT) are provided, and the spectral stability under long-term operation is not discussed.*

Our response: We thank the reviewer for this valuable comment. In response, we have added the Correlated Color Temperature (CCT) values for the red/white-light-emitting device in Supplementary Table 4. Regarding the spectral stability, as shown in Supplementary Fig. 16, the current devices suffer from very short operational lifetimes, which makes it difficult to perform meaningful long-term stability measurements at this stage.

Comment 5: *In methods, the authors should include more details in the Methods section regarding device encapsulation and measurement conditions.*

Our response: We thank the reviewer for this helpful suggestion. In the revised manuscript, we have added further details about the encapsulation procedure, measurement conditions, and device areas. Specifically, we now state that the encapsulation included the use of a desiccant in addition to a UV-epoxy resin and glass cover. For the photovoltaic measurements, we clarified that the light intensity was calibrated using a standard silicon photodiode. We also specified that the device areas were $3 \times 3 \text{ mm}^2$. The revised sentences in the *Methods* section now read as follows:

Original	The devices were encapsulated using a UV-epoxy resin in a nitrogen atmosphere after cathode formation.
----------	--

Corrected	The devices were encapsulated using a UV-epoxy resin, a glass cover, and a desiccant in a nitrogen atmosphere after cathode formation.
Original	The cells were illuminated with a spectrally mismatch-corrected intensity of 100 mW cm ⁻² (AM1.5G) provided by a sun simulator (OTENTO-SUNIIIIFNS, Bunkokeiki Co., Ltd.)
Corrected	The cells were illuminated with a spectrally mismatch-corrected intensity of 100 mW cm ⁻² (AM1.5G) provided by a sun simulator (OTENTO-SUNIIIIFNS, Bunkokeiki Co., Ltd.), with intensity calibration via a standard silicon photodiode. The device areas were 3 × 3 mm ² .

Comment 6: *To improve the clarity and conciseness of the main text, I recommend moving some of the redundant or less critical figures to the Supporting Information. This would help streamline the presentation and allow the key results to stand out more clearly.*

Our response: We appreciate the reviewer’s helpful suggestion. In the revised manuscript, we have streamlined the presentation by moving the J–V, L–V, and EQE–J characteristics that were previously included in Fig. 4a–c to the Supporting Information (now shown in Supplementary Fig. 7a–c). The main text now highlights only the key result that v-DABNA-based MF devices with various acceptors exhibited diverse emission colours (Fig. 4a). We then refer to the supplementary figures for the J–V–L and EQE_{EL} data, and continue to discuss the impact of acceptor choice on device performance and emission characteristics. The manuscript has been changed as follows (page 12):

Original	Figure 4 presents the characteristics of MF devices fabricated using v-DABNA, a representative blue-light-emitting MR-TADF donor with different acceptors. As shown in Figs. 4a–c, the J–V–L characteristics and EQE _{EL} exhibited values similar to those of MF devices with CzDBA as the acceptor. Additionally, a significant decrease in EQE _{EL} was observed in devices fabricated using acceptors with a lower E _T , such as rubrene and DBzA (Supplementary Fig. 1b). As depicted in Fig. 4d, EL spectra spanning from blue to red were obtained. A particularly noteworthy observation is the differences in EL spectra...
Corrected	When v-DABNA was used as the donor with various acceptors, the devices exhibited a wide range of emission colours (Fig. 4a). As shown in Supplementary Figs. 7a–c, the J–V–L characteristics and EQE_{EL} exhibited values similar to those of MF devices with CzDBA as the acceptor. Additionally, a significant decrease in EQE_{EL} was observed in devices fabricated using acceptors with a lower E_T, such as rubrene and DBzA (Supplementary Fig. 1b). A particularly noteworthy observation is the differences in EL spectra...

Reviewer #2

General comment: *In this manuscript, the authors report the simultaneous achievement of relatively high electroluminescence (EL) efficiency and power conversion efficiency (PCE) in a single organic device by employing MR-TADF materials. The authors demonstrated that precise modulation of E_b can regulate both the efficiency and emission color of the devices. This work is of potential interest to the OLED and OPV communities, as it expands the functionality of organic semiconductors toward self-powered displays and lighting applications. I would recommend acceptance of this manuscript after following issues are addressed.*

Our response: We are grateful to the reviewer for recommending publication of the manuscript in Nature Communications after some revisions. Changes are shown by red-letter in the revised manuscript. (The Supporting Information is written in black letter.)

Comment 1: *Donor- or acceptor-isolated EL spectra should be provided to distinguish emissions originating from CT states or TTA-UC. It is also recommended to include the absorption and PL spectra of the relevant donor and acceptor, eg. CzBDA. Please include the PL spectra of the five devices, particularly for the mixed films. In addition, provide L_{max} , PE_{max} , and CE_{max} for the devices.*

Our response: We thank the reviewer for this important comment. To clarify the origin of the emission, we have added additional spectroscopic data. Specifically, we included the PL and absorption spectra of a CzDBA neat film in Supplementary Fig. 2a to demonstrate that the EL spectra in Fig. 3 of MF devices other than those using rubrene are indeed CT-state-derived. In addition, because Cz-TRZ-Py is a material originally developed in our group and reported previously, its neat-film PL and absorption spectra were also added in Supplementary Fig. 2a. For the other donor/acceptor materials, the corresponding neat-film PL spectra are available in the cited references.

The manuscript has been changed as follows (pages 8-9):

Original	Given that CzDBA has an emission peak of around 550 nm, the observed EL spectra above 600 nm are attributed to emission from the CT state. ⁸⁻¹³
Corrected	Given that CzDBA has an emission peak of around 550 nm, the observed EL spectra above 600 nm are attributed to emission from the CT state (Supplementary Fig. 2a). ⁸⁻¹³

Regarding the PL spectra of the devices, since the blue emission originates from DABNA-2, we have provided the PL spectra of the three mixed films corresponding to the green-, yellow-, and orange-emitting devices in Supplementary Fig. 14. As now noted in the main text, these results confirm that the green, yellow, and orange EL emissions originate from CT states (Supplementary Fig. 14). The following discussion has been added to page 15:

To further clarify the emission origin, the PL spectra of the three mixed films corresponding to the green-, yellow-, and orange-emitting devices are shown in Supplementary Fig. 14. These results confirm that the green, yellow, and orange EL emissions mainly originate from CT states.

Finally, we have added L_{max} , PE_{max} , and CE_{max} for the devices in Supplementary Table 4. For clarity, the photovoltaic performances that were previously listed in Supplementary Table 4 have been separated and moved to Supplementary Table 5.

Comment 2: *The authors attributed the variation in donor emission intensity observed when employing different MR-TADF emitters (V-DABNA and DABNA-2) as donors to exciton dissociation induced by a low E_b , while it should be noted that differences in the carrier mobilities and HOMO energy levels of the donors can lead to variations in carrier transport within the devices, thereby affecting the location of the recombination zone, which implies that the shift of the recombination zone might be responsible for the change in donor EL emission intensity. Therefore, it is necessary to exclude the influence of donor-induced*

differences in carrier transport in order to further substantiate that the change in E_b is the factor responsible for the variation in donor EL emission intensity.

Our response: We thank the reviewer for raising this important point. To further investigate the possible influence of carrier injection and transport on the EL spectra, we compared four MF devices using v-DABNA or DABNA-2 as donors and B4PyMPM or Cz-TRZ-Py as acceptors (Supplementary Fig. 13). The results suggest that while differences in hole injection and transport between v-DABNA and DABNA-2 may exert some influence on exciton dissociation, this effect is less significant than that of E_b . Specifically, in devices employing v-DABNA as the donor, the smaller IE of v-DABNA led to higher current density and earlier hole arrival at the donor/acceptor interface. Such a situation would normally be expected to suppress donor emission due to enhanced hole flux at the interface. However, strong donor emission was observed only in devices with B4PyMPM, which yielded smaller E_b values. These findings indicate that, for emission-colour modulation in MF devices, the influence of E_b outweighs that of donor-dependent carrier injection and transport.

The following sentences have been added on page 15 of the revised manuscript:

To further examine the possible influence of carrier injection and transport on the EL spectra, we compared four MF devices using v-DABNA or DABNA-2 as donors and B4PyMPM or Cz-TRZ-Py as acceptors (Supplementary Fig. 13). The results suggest that differences in hole-injection and transport between v-DABNA and DABNA-2 may have some influence on exciton dissociation, but the effect appears less significant than that of E_b . In the devices employing v-DABNA as the donor, the smaller IE of v-DABNA led to higher current density and earlier hole arrival at the donor/acceptor interface. In such a situation, increased hole flux to the interface would generally be expected to suppress donor emission; however, strong donor emission was observed specifically in the devices with B4PyMPM, which yielded smaller E_b values. These results indicate that, for emission-colour modulation in MF devices, the influence of E_b outweighs that of donor-dependent carrier injection and transport.

Comment 3: *The authors discussed in detail the interfacial CT-state emission of organic MF devices in the manuscript. The modulation of CT-state emission is an effective strategy for improving EL devices, some related works (Adv. Mater. 2016, 28, 239–244; Adv. Mater. 2016, 28, 6758–6765; Adv. Mater. Interfaces 2018, 5, 1800025; Adv. Mater. 2024, 36, 2403584) could be included as references.*

Our response: We thank the reviewer for this helpful suggestion. We have revised the manuscript to include the relevant references on CT-state emission modulation (Adv. Mater. 2016, 28, 239–244; Adv. Mater. 2016, 28, 6758–6765; Adv. Mater. Interfaces 2018, 5, 1800025; Adv. Mater. 2024, 36, 2403584) as references 10–13. The sentence in pages 3 and 8 now reads:

“In OLEDs, charge injection and transport lead to exciton/CT state formation, followed by radiative recombination that results in light emission.”^{8–13}

“Given that CzDBA has an emission peak of around 550 nm, the observed EL spectra above 600 nm are attributed to emission from the CT state.”^{8–13}

Comment 4: *Although the authors achieved both high EL EQE and PCE in a single MF organic device, the performance of MF devices with emission colors ranging from blue to red, and even white, still exhibits a trend in which PCE decreases as EL EQE increases. How the authors view this issue, along with a brief discussion and outlook, may help guide further improvements in the performance of MF devices.*

Our response: We thank the reviewer for this constructive comment. We agree that the photovoltaic efficiency of the present MF devices is limited, and we have therefore included additional discussion to address the factors responsible for the relatively low PCE_{PV} and possible strategies for improvement. In particular, we describe how, in MF devices incorporating a mixed film of v-DABNA and CzDBA, increasing the film thickness results in a higher J_{sc} but a reduced FF

due to the large offset between the IE and EA of the two materials (Supplementary Fig. 17). We also discuss results from MF devices employing HN-D2 together with DikTA, which has a complementary absorption range, showing that such a configuration enhances J_{sc} in a manner analogous to conventional bulk-heterojunction OPVs (Supplementary Fig. 18). Based on these results, we suggest that the use of mixed donor/acceptor layers with complementary absorption spectra, together with carefully designed energy levels that facilitate charge transport, provides a pathway to improving PCE_{PV} .

The following discussion has been added before Conclusion section:

In this study, although the PCE_{PV} was limited to a maximum of ~1.5%, we obtained several indications that point toward possible strategies for improvement. For example, in MF devices incorporating a mixed film of v-DABNA and CzDBA, increasing the mixed film thickness led to a higher J_{sc} but a reduced FF (Supplementary Fig. 17). This behaviour can be explained by the fact that a thicker mixed layer increases the number of charge-generating interfaces, while the decrease in FF is likely caused by a large offset between the IE and EA of v-DABNA and CzDBA. Furthermore, in MF devices employing HN-D2 together with DikTA, which possesses a complementary absorption range, we observed an enhancement in J_{sc} similar to that typically seen in bulk-heterojunction OPVs (Supplementary Fig. 18).⁴³ These results suggest that the use of mixed layers containing donor/acceptor pairs with complementary absorption spectra, combined with carefully designed energy levels that do not hinder charge transport, can provide a pathway to further improvement in PCE_{PV} .

Comment 5: *In Fig. S2b, please provide the PLQY for the two materials. This will allow assessment of the effect of E_b on the donor emission intensity when controlling the variable of EA. In the statistical analysis of E_b and E_{cs} in Fig. S6, please clarify the function used. In Fig. S7, is E_b positively correlated with PLQY? Are there other influencing factors? Please elaborate.*

Our response: We thank the reviewer for these constructive comments. In response, we have made the following revisions:

We have added the PLQY values of the two neat films in Fig. S2b to allow assessment of the relationship between E_b and donor emission intensity when EA is controlled.

In Fig. S8, we added a fitted line (least-squares method) to the plot of E_b of the CT state as a function of E_{cs} . The slope was approximately 0.3, which is intriguingly similar to the slope (0.25) reported for isolated molecules (not CT states). Since reports on E_b of CT states are still limited, we plan to acquire additional data to enable more detailed analysis in the future. The following discussion has been added to page 12:

For all the CT states examined so far, the least-squares fitting of E_b as a function of E_{cs} yielded a slope of approximately 0.3, which is intriguingly similar to the slope of 0.25 reported for isolated molecules.³⁰ Because reports on the E_b of CT states are still scarce, we anticipate that acquiring more data will allow a more detailed analysis in the future.

In Fig. S9, the description in the main text has been revised to clearly indicate a positive correlation between E_b and the PLQY of exciplex emission. We also noted the potential influence of the donor and acceptor PLQY on exciplex PLQY, but emphasized that the current dataset is limited and that further data will be required to confirm this effect. The manuscript has been changed as follows (pages 12-13):

Original	Furthermore, in donor–acceptor blend films, a lower E_b was found to result in a lower PLQY of exciplex emission (Supplementary Fig. 7).
----------	--

Corrected	Furthermore, in donor–acceptor blend films, a positive correlation was observed between E_b and the PLQY of exciplex emission (Supplementary Fig. 9). While the PLQY of the individual donor and acceptor may also influence the exciplex PLQY, the current dataset is limited, and further data collection will be required to clarify these potential effects.
-----------	--

Comment 6: For Fig. 5, it is suggested to supplement with an energy-level diagram of the device, including E_b , E_T , and ECS for both the donor and acceptor, to enable more intuitive comparative analysis for the reader.

Our response: We thank the reviewer for this constructive suggestion. Following the recommendation, we have supplemented the manuscript with an energy-level diagram (Figure S12) that includes E_b , E_T , and ECS for both donors and acceptors, where such information could be reliably obtained. However, in the case of the mixed film comprising DtBuCzB and tBuCzDBA (orange), three types of emission—donor, acceptor, and exciplex—were observed in the EL and PL spectra (Figure S14b). As a result, reliable estimation of E_T and E_b for this system was not feasible.

Comment 7: How does the interfacial morphology of the films affect device performance? How does the carrier mobility of different donor and acceptor materials influence the performance? In the discussion of results, it is recommended to summarize the effects of donor E_T , exciplex E_{CT} , ECS, E_b , and PLQY on device EQE and PCE, in order to guide material selection and performance trade-offs in practical applications.

Our response: We thank the reviewer for this valuable comment. We agree that the effects of interfacial morphology, carrier mobility, and various energy parameters on device performance are critical issues for understanding and improving MF devices. At present, it is difficult to directly investigate the influence of interfacial morphology in the current device systems; however, we plan to explore this in future work by employing bulkier acceptors such as iCzDBA, which has a molecular structure similar to that of tBuCzDBA but with increased steric bulk (see: <https://doi.org/10.1039/D2SC04725J>). This will allow us to assess how molecular steric effects at the donor–acceptor interface impact device characteristics.

Regarding carrier mobility, we have revised the manuscript to clarify that although the electron mobility of a tBuCzDBA neat film ($\sim 10^{-6} \text{ cm}^2 \text{ V}^{-1} \text{ s}^{-1}$) is an order of magnitude lower than that of a CzDBA neat film ($\sim 10^{-5} \text{ cm}^2 \text{ V}^{-1} \text{ s}^{-1}$), the PCE_{PV} values of v-DABNA-based MF devices were comparable in both cases, suggesting that such a difference in electron mobility has only a minor impact on PCE_{PV} . However, the OLED J – V characteristics revealed that devices employing tBuCzDBA required a higher operating voltage, indicating that transport properties still influence the EL behaviour.

Furthermore, we have added a summary of how donor E_T , exciplex E_{CT} , ECS, E_b , and PLQY affect device EQE and PCE. In particular, we now highlight that while no clear correlation was found between E_b and PCE_{PV} , a larger E_{CT} tends to result in a higher PCE_{PV} , consistent with the increase in V_{OC} observed for larger E_{CT} values. We would also like to note that the influence of E_b on PLQY and device EQE has already been addressed in the manuscript: specifically, Fig. 4c shows that devices with lower E_b exhibit lower EQE_{EL} , indicating that a threshold E_b is required for efficient CT-state emission. Moreover, Supplementary Fig. 9 demonstrates a positive correlation between E_b and the PLQY of exciplex emission, although additional data will be needed to fully clarify these effects. This discussion should help guide material selection and trade-off considerations in practical applications of MF devices.

The manuscript has been changed as follows (pages 13–14):

Original	This can be attributed to the simple donor/acceptor stacked structure of the devices, as well as the relatively high mobility of materials such as CzDBA, which was used as the
----------	---

	acceptor as shown in Fig. 3. ⁴⁵
Corrected	This can be attributed to the simple donor/acceptor stacked structure of the devices. The electron mobility of a tBuCzDBA neat film is on the order of $10^{-6} \text{ cm}^2 \text{ V}^{-1} \text{ s}^{-1}$, whereas that of a CzDBA neat film is on the order of $10^{-5} \text{ cm}^2 \text{ V}^{-1} \text{ s}^{-1}$. Nevertheless, the PCE_{PV} of v-DABNA-based MF devices was comparable in both cases, suggesting that a one-order difference in electron mobility exerts only a minor influence on PCE_{PV} (Figs. 4e and 4f). ^{49,50} In contrast, in the OLED characteristics (J - V curves) of the v-DABNA-based MF devices, those employing tBuCzDBA required a higher operating voltage (Fig. 3a and Supplementary Fig. 7a). Although no clear correlation was observed between E_{b} and PCE_{PV} (Fig. 4f), a trend was identified in which a larger E_{CT} led to a higher PCE_{PV} (Supplementary Fig. 11a). This can be attributed to the fact that an increased E_{CT} results in a higher V_{OC} (Supplementary Fig. 11b).

Comment 8: *The caption “Figure 3f shows the correlation between the maximum EQEEL and E_{b} ” should be corrected to “Figure 4f shows the correlation between the maximum EQEEL and E_{b} .”*

Our response: Thank you for pointing that out. We have revised the manuscript as follows (page 12).

Original	Figure 3f shows the correlation between the maximum EQE_{EL} and E_{b} in devices where exciplex emission was the dominant luminescence.
Corrected	Figure 4c shows the correlation between the maximum EQE_{EL} and E_{b} in devices where exciplex emission was the dominant luminescence.

Reviewer #3

General comment: *This manuscript reports on excellent and original work, discovering molecular donor:acceptor material systems with combining a high EL quantum efficiency and high photovoltaic efficiency. The work is important as it provides important insights into materials design rules allowing to combine a high free charge carrier generation yield upon illumination with a high LED quantum efficiency. such materials are key to achieve power conversion efficiencies close to the fundamental shockley-queisser limit. This work should be published and I only have a few comments which I hope the authors will take into account:*

Our response: We are grateful to the reviewer for recommending publication of the manuscript in Nature Communications after some revisions. Changes are shown by red-letter in the revised manuscript. (The Supporting Information is written in black letter.)

Comment 1: *First sentence of the abstract: “Achieving both high electroluminescence (EL) efficiency and power conversion efficiency (PCE) in a single organic device has long been considered challenging, as these processes are fundamentally opposite.”*

I would argue that these processes are not fundamentally opposite: In fact, the Shockley-Queisser limit for photovoltaics dictates that the highest PCE will be achieved for the absorber with the highest EL efficiency (since that one will have the highest Voc). I would ask the authors to reconsider or soften this statement. I would agree that it is difficult to achieve both high EL and high PV PCE as it involves reducing the binding energy of the interfacial states. But this is not fundamentally impossible.

Also from the abstract “In this study, we present a novel strategy for material selection and device design to overcome this limitation and realise the unprecedented coexistence of high EL and photovoltaic (PV) efficiencies.” Authors should mention already in the abstract that this new strategy is to make use of multiple-resonance thermally activated delayed fluorescence (MR-TADF) materials, which exhibit high emission efficiency and strong absorption.

Our response: We sincerely thank the reviewer for this valuable comment. Following the suggestion, we have revised the first sentence of the abstract to avoid the misleading expression “fundamentally opposite.” It now reads:

“...has long been considered difficult, since the design principles optimising one often compromise the other.”

This modification softens the statement, in agreement with the reviewer’s point that achieving both high EL and PCE is difficult but not fundamentally impossible.

In addition, as recommended, we have explicitly mentioned our use of multiple-resonance thermally activated delayed fluorescence materials in the abstract. The revised sentence now reads:

“...we present a novel strategy employing multiple-resonance thermally activated delayed fluorescence materials with strong absorption and high emission efficiency, enabling coexistence of high EL and photovoltaic (PV) efficiencies.” The abstract has been changed as follows:

Original	Achieving both high electroluminescence (EL) efficiency and power conversion efficiency (PCE) in a single organic device has long been considered challenging, as these processes are fundamentally opposite. In this study, we present a novel strategy for material selection and device design to overcome this limitation and realise the unprecedented coexistence of high EL and photovoltaic (PV) efficiencies.
Corrected	Achieving both high electroluminescence (EL) efficiency and power conversion efficiency (PCE) in a single organic device has long been considered difficult, since the design principles optimising one often compromise the other. In this study, we present a novel strategy employing multiple-resonance thermally activated delayed fluorescence materials with strong absorption and high emission efficiency, enabling coexistence of high EL and photovoltaic (PV) efficiencies.

Comment 2: *To avoid confusion, throughout the manuscript, authors should clarify when they mention PCE, that it is photovoltaic PCE (PCE_{PV}) and not LED PCE (electron to photon power conversion efficiency). When they mention EQE, they should clarify when it is LED EQE (electron-to-photon) or photovoltaic EQE (photon-to-electron).*

Our response: We thank the reviewer for this helpful suggestion. To avoid confusion, we have revised the manuscript to consistently use PCE_{PV} instead of PCE. In addition, we have clarified its definition in the Introduction as follows (page 4):

“...power conversion efficiency (PCE_{PV}, defined as the ratio of generated electrical power to incident optical power in photovoltaic operation)...”

We also note that EQE_{EL} (external quantum efficiency of electroluminescence) is explicitly defined on page 4, making it clearly distinct from photovoltaic efficiency metrics. Furthermore, in response to Comment 6, we introduced photovoltaic external quantum efficiency data; however, in this case we deliberately avoided using the abbreviation “EQE” to prevent confusion with EQE_{EL}.

Comment 3: *Introduction: “Furthermore, if both EL and PV functions can be combined, it would be possible to realise a mobile display that can be charged using indoor lighting or sunlight when not in use. This would free people from the hassle of charging and significantly reduce the overall power consumption of displays in society.” This statement is partially true, as for the best performance, the PV device should of course absorb part the NIR of the sunlight, which is not possible if it also has to emit visible light.*

Our response: We appreciate the reviewer’s insightful comment. We have revised the Introduction to moderate our statement and to acknowledge the trade-off between absorption in the near-infrared region, which is beneficial for PV operation, and visible emission required for EL. The manuscript has been changed as follows (page 4):

Original	Furthermore, if both EL and PV functions can be combined, it would be possible to realise a mobile display that can be charged using indoor lighting or sunlight when not in use. This would free people from the hassle of charging and significantly reduce the overall power consumption of displays in society
Corrected	Furthermore, if both EL and PV functions can be combined, it could enable mobile displays that recharge under indoor lighting or sunlight when not in use. While we note that the optimal PV device should ideally harvest part of the near-infrared spectrum of sunlight, which is difficult to exploit simultaneously for visible emission, the demonstration of efficient MF devices nonetheless suggests a pathway toward reducing charging needs and lowering the overall energy consumption of displays.

Comment 4: *Introduction: “Thus, in organic-semiconductor-based devices, it is inherently challenging to achieve high performance in both EL and PV functions, which are fundamentally opposite processes.” See my previous comment: if non-radiative pathways are minimized, the photovoltage losses and EQE_{EL} will be simultaneously maximized. They are thus not fundamentally opposite. Decreasing the binding energy might increase the change for the charge carriers to find non-radiative pathways, but it is not a fundamental limitation if non-radiative pathways are minimized.*

Our response: Thank you for pointing that out. In addition to revising the abstract, we have also modified the corresponding sentence in the *Introduction* to soften the expression.

Original	Thus, in organic-semiconductor-based devices, it is inherently challenging to achieve high performance in both EL and PV functions, which are fundamentally opposite processes.
Corrected	Thus, in organic-semiconductor-based devices, it is inherently challenging to achieve high performance in both EL and PV functions, as the design principles that enhance one often compromise the other.

Comment 5: *Results and discussion: “The EL characteristics shown in Fig. 3a indicate that all devices exhibit current flow at applied voltages below 2 V, reaching a current density of approximately 100 mA/cm² at around 4 V. This observation suggests an efficient charge injection at the electrode/organic semiconductor interface.” The low onset voltage for emission is the result of a low excited state binding energy, see reference 19. Even if charge injection and the electrode/organic interface would be efficient, it is still necessary to reduce the CT state binding energy in order to achieve such low onset voltages.*

Our response: We thank the reviewer for this valuable comment. We agree that the low onset voltage for emission is not solely attributable to efficient charge injection, but also to the reduction of the CT state binding energy, as reported in ref. 19. The manuscript has been changed as follows (page 8):

Original	This observation suggests an efficient charge injection at the electrode/organic semiconductor interface.
Corrected	This observation suggests an efficient charge injection at the electrode/organic semiconductor interface, while the low onset voltage for emission can also be attributed to the small E_b of the CT state, consistent with previous reports ²³ . As discussed later, many of the devices investigated in this study indeed involve donor–acceptor pairs with relatively small CT-state E_b , which further supports the observed low-voltage EL operation.

Comment 6: *Absorption spectra, or more ideally, photovoltaic external quantum efficiency spectra, are missing from the main text. They would corroborate the statement that MR-TADF materials are strong absorbers.*

Our response: We thank the reviewer for pointing that out. In response, we have added the photovoltaic external quantum efficiency spectra of two MF devices employing MR-TADF donors to Supplementary Fig. 5. For example, the device combining DABNA-2 and B4PyMPM exhibited an EQE_{PV} of ~25%, which is comparable to the performance of recently reported OPDs utilizing MR-TADF materials. This addition supports our statement that MR-TADF materials act as strong absorbers in MF devices. The following discussion has been added to pages 10 and 25:

The photovoltaic external quantum efficiency of devices employing MR-TADF materials as donors reached approximately 25%, which is comparable to the values reported for state-of-the-art OPDs based on MR-TADF materials (Supplementary Fig. 5).³⁴

Photovoltaic external quantum efficiency and response spectra of the devices were collected using a Spectral sensitivity measurement system VC-250 (Bunkokeiki Co., Ltd.) under zero-bias conditions.

Our Responses to the Comments of the Reviewers

Reviewer #1

Comment 1: *The only minor suggestion is that the authors could briefly comment in the Discussion section on the potential stability challenges of MR-TADF materials under simultaneous EL and PV operation, as Supplementary Fig. S16 indicates limited device lifetime.*

Our response: We thank the reviewer for this constructive suggestion. We have added a brief discussion in the revised manuscript describing the stability characteristics of the blue and yellow MF devices under EL and PV operation. Specifically, we note that the blue device exhibited a shorter EL lifetime but a longer PV lifetime than the yellow device. This highlights the trade-off between high efficiency and operational stability, which requires further material development to overcome. The following sentences have been added (page 17):

The blue device exhibited a shorter EL lifetime but a longer PV lifetime than the yellow device. Although both devices showed comparable PCE_{PV} values, the higher EL efficiency of the yellow device was accompanied by a shorter PV lifetime, suggesting that achieving both high efficiency and long-term stability remains challenging. Continued material development will be essential to address this trade-off.

Reviewer #2

Comment 1: *The authors have demonstrated multifunctional devices with emission colors ranging from blue to red and even white. However, there are significant performance disparities, particularly in EQEEL and PCEPV, among the devices of different colors (e.g., blue/red EQEEL; 2% vs. green/orange; 8.5%). It is recommended that the authors provide a deeper analysis of the fundamental reasons behind these differences. Are they related to the intrinsic PLQY of the different MR-TADF materials, variations in charge carrier mobility, or differences in non-radiative decay pathways of the charge-transfer states formed with the respective acceptors?*

Our response: We thank the reviewer for this valuable comment. We agree that the performance variations among devices of different emission colours stem from both intrinsic material properties and device structural factors. To address this point, we have revised the manuscript to discuss the role of the doped layers used in the green- and orange-emitting MF devices. These doped layers exhibited higher PLQY than undoped films, and the expanded recombination region achieved through the introduction of the doped layer further enhanced EQE_{EL}. This combined effect of improved radiative efficiency and a broadened recombination region accounts for the superior EQE_{EL} observed in the green and orange devices compared with the blue and red devices. We have therefore added the following sentences (page 16):

These devices employed doped layers that exhibited higher PLQY than the corresponding undoped films, and the expansion of the recombination region achieved through the introduction of the doped layer further enhanced EQE_{EL}. This combination of improved radiative efficiency and a broadened recombination region likely accounts for the superior EQE_{EL} of the green and orange devices relative to the blue and red devices.

Comment 2-1: *The authors demonstrated that the exciton E_b is a key parameter for tuning both emission color and device efficiency, attributing its reduction to the multi-fused-ring structure and steric hindrance of the MR-TADF materials. It is recommended that the authors further discuss whether this strategy of modulating E_b through molecular structure implies a universal design rule. For instance, can all donors with large, planar, and multi-fused ring structures effectively lower E_b? Furthermore, is there an optimal range for E_b that best balances high EQEEL and high PCEPV? The current conclusion suggests that "E_b cannot be too low (otherwise CT-state radiative recombination efficiency suffers), but its impact on PV efficiency is relatively minor." Could this conclusion be elaborated more quantitatively?*

Our response: We thank the reviewer for this insightful comment. As suggested, we have expanded the discussion on the structural factors that influence E_b and their potential as design parameters for future molecular engineering. Specifically, although it is difficult to quantitatively correlate the number of fused rings with E_b because the MR-TADF donors used in this study exhibit different fused-ring geometries, we have clearly observed that steric hindrance has a significant influence on E_b. We have therefore added the following sentences (page 10):

Original	Since MR-TADF materials and HN-D2 have a greater number of fused rings than typical arylamine-based donors, it is considered that the E _b of the CT state was reduced. A comparison between indolo[3,2,1-de]indolo[3',2',1':8,1][1,4]benzazaborino[2,3,4-kl]phenazaborine (CzBN) and 2,6-bis(3,6-di-tert-butyl-9H-carbazol-9-yl)boron (DtBuCzB) suggests that the steric hindrance of the tertial butyl group is also effective in reducing E _b .
Corrected	Since MR-TADF materials and HN-D2 have a greater number of fused rings than typical arylamine-based donors, it is considered that the E _b of the CT state was reduced. Although it is difficult to quantitatively discuss the effect of the number of fused rings on E _b because the MR-TADF donors used in this study possess different fused-ring geometries, a clear influence of steric hindrance has been observed. A comparison between

indolo[3,2,1-de]indolo[3',2',1':8,1][1,4]benzazaborino[2,3,4-kl]phenazaborine (CzBN) and 2,6-bis(3,6-di-tert-butyl-9H-carbazol-9-yl)boron (DtBuCzB) suggests that the steric hindrance of the tertial butyl group is also effective in reducing E_b . Furthermore, a noticeable difference in E_b was also identified between 5,9-Diphenyl-5,9-diaza-13b-boranaphtho[3,2,1-de]anthracene (DABNA-1) and DABNA-2, which likely originates from the different degrees of steric hindrance around their boron centres.
--

Comment 2-2: *Furthermore, is there an optimal range for E_b that best balances high EQEEL and high PCEPV? The current conclusion suggests that " E_b cannot be too low (otherwise CT-state radiative recombination efficiency suffers), but its impact on PV efficiency is relatively minor." Could this conclusion be elaborated more quantitatively?*

Our response: We appreciate the reviewer's thoughtful suggestion. We agree that identifying an optimal range of E_b is important for achieving a balance between high EQEEL and high PCEPV. At present, it is difficult to quantitatively determine this range because the present study mainly focuses on simple donor/acceptor-stacked structures to verify the fundamental operating principle of MR-TADF-based MF devices. However, as shown in Fig. 5, the use of doped films resulted in significantly higher EQEEL, indicating that E_b optimization in doped-film-based MF devices will be a key step toward realising both high EQEEL and high PCEPV. Accordingly, we have added the following sentence before the Conclusion section (page 17-18):

To achieve a balance between high EQEEL and PCEPV, a deeper understanding of the influence of E_b is also crucial. Although this study mainly focused on simple donor/acceptor-stacked structures, further clarification of how E_b affects device characteristics in doped-film-based MF devices will be essential for identifying its optimal range.